# *Limnophila aromatica* Crude Extracts as Natural Emulsifiers for Formation and Stabilizing of Oil-in-Water (O/W) Emulsions

Rasmey Soeung [1,2], Lorena de Oliveira Felipe [1], Meryem Bouhoute [3], Noamane Taarji [4], Mitsutoshi Nakajima [5] and Marcos A. Neves [5,*]

1 Graduate School of Life and Environmental Sciences, University of Tsukuba, Tsukuba 305-8577, Japan; rasmeysoeung.rua@gmail.com (R.S.); lorenaob@gmail.com (L.d.O.F.)
2 Faculty of Agro-Industry, Royal University of Agriculture, Dangkor District, Phnom Penh P.O. Box 2696, Cambodia
3 Alliance for Research on the Mediterranean and North Africa (ARENA), University of Tsukuba, Tsukuba 305-0006, Japan; bouhoute.meryem.fu@u.tsukuba.ac.jp
4 Biodiversity and Plant Sciences Program, Mohammed 6 Polytechnic University (UM6P), AgroBioScience, Benguerir 43150, Morocco; noamane.taarji@um6p.ma
5 Faculty of Life and Environmental Sciences, University of Tsukuba, Tsukuba 305-8577, Japan; nakajima.m.fu@u.tsukuba.ac.jp
* Correspondence: marcos.neves.ga@u.tsukuba.ac.jp

**Abstract:** This study mainly focused on the emulsifying performance of *Limnophila aromatica* crude extracts obtained by using different ethanolic aqueous solutions (0, 25, 50, 75, and 99.5% ($v/v$)). All *Limnophila aromatica* extracts (LAEs) were able to produce emulsions with a volume mean droplet diameter ($d_{4,3}$) ranging from 273 to 747 nm, except for LAE-99.5 (3269 nm). Only the emulsion prepared by LAE-75 was stable during seven days of storage, without significantly changing droplet size (479–495 nm). The result showed that all LAEs could reduce interfacial tension varied within 12.5 and 16.1 mN/m at the soybean oil/extracts (1% $w/w$) interface. Compared to other extracts, LAE-75 did not contain the highest protein, saponin, and phenol content (4.36%, 20.14%, and 11.68%, respectively), but it had the lowest ash content (14.74%). These results indicated that the emulsifying performance of LAEs did not rely only on interfacial tension and/or surface-active compounds. The residual demulsifiers, such as inorganic substances, were also significantly involved in the emulsions' destabilization. Finally, the emulsion consisting of 0.5% ($w/w$) LAE-75 and 5% ($w/w$) soybean oil showed considerable stability during storage up to 30 days at different temperatures (5 or 25 °C). Therefore, *Limnophila aromatica* extract has a potential application as a new source of natural emulsifier.

**Keywords:** *Limnophila aromatica*; interfacial tension; oil-in-water (O/W) emulsion; protein; saponin; phenolic compound

## 1. Introduction

Emulsifiers are the essential substances used for forming and stabilizing emulsions in various industries, including the cosmetics, food, and pharmaceutical industries [1]. Most of the utilized emulsifiers are derived from chemical and/or enzymatic reactions [2]. Currently, food labels with terms such as "natural" or "sustainable" may have a considerable impact on the desirability and perceived quality of products [3]. Therefore, replacing the synthetic emulsifiers with natural emulsifiers is particularly interesting to meet the consumer demands on green label products. The main natural emulsifiers, such as polysaccharides, phospholipids, and proteins, are successfully used for emulsion-based products' preparation. Nevertheless, emulsions stabilized by these substances are possibly disrupted [4,5]. Hence, the developments of natural emulsifiers from other surface-active elements, such as saponins, are typically investigated to improve emulsifying activities [6,7].

Saponins are small-molecular-weight substances found in over 100 plant families and a few marine sources [8]. Plant species are the main factor affecting the composition and concentration of saponin extracts. However, the quality and quantity of saponin extracts also depend on seasonal changes, parts of the plant, and extraction conditions [9,10]. Some studies reported that saponins extracted from plant materials contained a mixture of different saponin compounds (i.e., more than 100 saponins obtained from *Quillaja saponaria* Molina extraction) and the residue of plant substances, resulting in impurity saponins [11,12]. Saponins have an amphiphilic nature comprising carbohydrate chains that act as hydrophilic links to steroid or triterpenoid aglycon, which acts as a hydrophobic. The mixture of these polar and non-polar structural compounds in their molecules describes the dispersion abilities of saponins [13,14]. Therefore, utilizations of the saponins (mostly quillaja saponins) as natural emulsifiers have attracted the attention of researchers. The emulsions formed by quillaja saponins were reported to have a long shelf-life and stability against environmental stresses such as heat treatment, pH change, ionic strength, and storage time [6,15–17].

Investigating various botanical sources of saponins has become an important area of research due to the increasing demand for using saponin extracts as natural emulsifiers. In recent years, different crude saponin-rich plant extracts from yucca (*Yucca schidigera*), ginseng (*Panax ginseng*), red beet (*Beta vulgaris*), oat bran (*Avena sativa* L.), argan press cake (*Argania spinosa* L.), sugar beet (*Beta vulgaris*), argan shells (*Argania spinosa* L.), olive oil cake, and liquorice root (*Glycyrrhiza glabra*) were successfully used as natural emulsifiers for preparing oil-in-water (O/W) emulsions. The saponin contents of the yucca, ginseng, red beet, oat bran, argan press cake, sugar beet, argan shells, olive oil cake, and liquorice root extracts were reported to be 9.5%, 7%, 0.9%, 4.6%, 4.2%, 0.5%, 10.4–39.1%, 7.8–9.2%, and 7.9–12.7%, respectively. Although saponin concentrations of those extracts were relatively low, their performances in emulsion formation were comparable to quillaja saponin [18–26]. It indicates that other substances, such as phenolic compounds and proteins, also make significant contributions to the emulsifying activities of the crude extracts. The researchers generally selected the specific extract for preparing the emulsion based on its ability to decrease the interfacial tension and/or the high surface-active compounds [20,22,23]. However, the existence of residual particles in the crude extracts could significantly influence their physicochemical properties, leading to emulsion instability [26]. Therefore, it is necessary to compare the emulsions formulated by different extracts obtained from the same material with varying extraction conditions.

*Limnophila aromatica* (rice paddy herb) is a tropical aromatic herb that belongs to the Scrophulariaceae family [27]. The plant often grows in a hot and watery environment, mainly in flooded rice fields. It originated from Southeast Asia and is commonly used as spices and medicinal herbs [28]. Rice paddy herb is traditionally used to cure convulsion, anxiety, and stress; protect against vascular dysfunction; treat fever; and give to nursing women. It is known as a major ingredient for fresh fish dishes because of the flavor and aroma reminiscent of both lemon and cumin [29,30]. The extracts of *Limnophila aromatica* were reported to contain anti-inflammatory and antioxidants effects [31,32]. Due to its unique scent, essential-oil extractions from *Limnophila aromatica* were commonly conducted, and methyl benzoate, pulegone, limonene, z-ocimene, terpinolene, and camphor were found to be the major compounds [29,33,34]. Recently, the studies of this plant's phenol and flavonoid extraction and starch isolation were also reported. The investigation showed that total phenol (4%) and flavonoid (3.1%) of the extract were obtained from ethanolic extraction [35]. Wijaya and co-workers found that the defatted and dephenolated *Limnophila aromatica* contained starch with a purity of 70.4% (55.1% resistant starch) [36]. However, the contents of saponin, protein, and other substances of *Limnophila aromatica* related to the emulsifying performance of the extracts have not been reported in the previous studies.

This work aimed to investigate the overall emulsifying performance of rice paddy herb (*Limnophila aromatica*) crude extracts by different aqueous ethanolic extraction. First, the interfacial tension and the surface-active compositions, with particular interest paid

to phenolic, saponin, and protein, were evaluated. According to the significant impact of the residual substances on emulsions destabilization, the total ash content of the extracts was necessary to determine. The volume mean droplet diameter ($d_{4,3}$), electrical charge ($\zeta$-potential), and droplet size distribution were observed to evaluate the impact of the different extracts on the oil-in-water (O/W) emulsion formation and stabilization. The selected extract was then used to prepare the emulsions with various concentrations of extract and oil. Finally, the stability of the emulsion was also assessed by storage at two different temperatures (5 and 25 °C) for 30 days. This is the first report on the usage of rice paddy herb extracts as the natural emulsifiers for forming and stabilizing oil-in-water (O/W) emulsion.

## 2. Materials and Methods

### 2.1. Materials and Chemicals

*Limnophila aromatica* were obtained from a local farm in Takeo province, Cambodia. We purchased 99.5% ethanol ($C_2H_5OH$), gallic acid ($C_7H_6O_5$), Oleanolic acid ($C_{30}H_{48}O_3$), D (+)-glucose ($C_6H_{12}O_6$), vanillin ($C_8H_8O_3$), acetic acid ($CH_3COOH$), 60% perchloric acid ($HClO_4$), ethyl acetate ($CH_3COOC_2H_5$), phenol ($C_6H_5OH$), sulfuric acid ($H_2SO_4$), sodium carbonate ($Na_2CO_3$), sodium azide ($NaN_3$), and soybean oil from FUJIFILM Wako Pure Chemical Corporation (Osaka, Japan). Folin–Ciocâlteu (FC) reagent was acquired from Sigma-Aldrich (Tokyo, Japan). The ultrapure water utilized in the study was produced by Alium ® comfort II system (Sartorius AG, Göttingen, Germany).

### 2.2. Samples and Extracts Preparation

The aerial parts of *Limnophila aromatica* were washed with distilled water and air-dried for nearly three days. The dried plants were packed in zip plastic bags, transferred to the University of Tsukuba, Japan, and stored at −20 °C until further use. Before extraction, the dried samples were ground into homogeneous powder by the Ultra Centrifugal Mill ZM 200 at 6000 rpm with a particle size of less than 0.5 mm. The solvent extraction was carried out at room temperature for 3 h by stirring the *Limnophila aromatica* powder in different aqueous ethanolic solutions (0, 25, 50, 75, or 99.5% (*v/v*)) at a powder:solvent weight ratio of 1:10. The mixtures were then centrifuged at 2280× *g* for 30 min by Kubota Corp, Tokyo, Japan, and filtered through Whatman filter paper to separate the solid suspensions. To eliminate the solvent, the obtained supernatant was evaporated by using a rotary-evaporator EYELA Co., Ltd., Shanghai, China, at 49 hPa and 40 °C. The obtained extracts were then mixed with the water, centrifuged at 9100× *g* for 30 min, with high-speed refrigerated micro centrifuge MX-307, TOMY, Japan, and filtered by RephiQuick non-sterile syringe filter (PTFE 0.45 μm) to remove the insoluble substances. Finally, the filtered extracts were lyophilized at −80 °C, 5 Pa, using an EYELA freeze-drier, Shanghai, China, to eliminate the water. The freeze-dried water-soluble *Limnophila aromatica* extracts, namely LAE-0, LAE-25, LAE-50, LAE-75, and LAE-99.5, were stored at −20 °C until the further experimentation. The extraction yield (EY) was calculated as indicated in Equation (1):

$$EY\ (\%,\ dry\ basis) = W_1/W_0 \times 100 \qquad (1)$$

where $W_1$ represents the weight of the freeze-dried Limnophila aromatica extract, and $W_0$ represents the weight of Limnophila aromatica powder.

### 2.3. Physicochemical Characterization of Extracts

2.3.1. Chemical Composition Characterization

The saponin content was determined spectrophotometrically by using oleanolic acid as a standard (0–250 μg/mL) for preparing the calibration curve, $R^2 = 0.9957$. In brief, 0.1 mL of each extract (500 or 1500 μg/mL) was added into the test tube, followed by 0.1 mL of 5% vanillin–acetic acid solution and 1.2 mL of 60% perchloric acid. The mixture was then incubated for 20 min at 70 °C and cooled down at room temperature. Then 5 mL of ethyl acetate was added, and the absorbance of the mixture was immediately measured

at 550 nm by UV–VIS spectrophotometer (JASCO Co., Hachioji, Japan). The blank was made with the same procedure without the extract [37].

The total phenol content of each extract was determined following the Folin–Ciocâlteu method described by Sahu and Saxena [38]. Gallic acid was used as a standard (0–80 μg/mL) for establishing the calibration curve, $R^2$ = 0.9993. Briefly, 0.5 mL of each extract (250 or 500 μg/mL) was added into the test tube, followed by 2.5 mL of a 10-fold diluted Folin–Ciocâlteu reagent and 2 mL of 7.5% sodium carbonate and allowed to stand for 30 min at room temperature. The absorbance of the mixture was then read at 760 nm by UV–VIS spectrophotometer (JASCO Co., Hachioji, Japan). The blank was made with the same procedure, without the extract.

The total protein content of each extract was estimated by multiplying the total nitrogen content by 6.25 (the nitrogen to protein conversion factor) [39]. The total nitrogen content was measured by using an organic elemental analyzer (C, H, N, S) (elementar (UNICUBE)) by the Research Facility Center for Science and Technology, Chemical Analysis Division of the University of Tsukuba.

The ash content was estimated by burning off the organic matter. Briefly, 1–2 g of *Limnophila aromatica* extract was placed in the dried/pre-weighted porcelain crucible. The crucible containing sample was then put in the furnace muffle at 600 °C for 24 h and then set in the desiccator to cool to ambient temperature for about 30 min [40]. The crucible containing ash was weighed, and the ash content was calculated as indicated in Equation (2):

$$\text{Ash content (\%, dry basis)} = W_1/W_0 \times 100 \tag{2}$$

where $W_1$ represents the weight of the ash, and $W_0$ represents the weight of the freeze-dried *Limnophila aromatica* extract.

### 2.3.2. Interfacial Tension Measurements

Interfacial tensions of the solutions with a concentration of 0.005–3% (*w/w*) *Limnophila aromatica* extracts dissolved in ultrapure water were measured at 25 °C by the pendant drop method, using an interfacial tensiometer (DMo-501, Kyowa Interface Science Co., Ltd., Saitama, Japan) at soybean oil/water interface. In brief, a 22-gauge syringe needle (22 G) was used to inject the extract solution into the soybean oil. An image of the drop was captured immediately by a high-resolution camera after reaching the maximum drop volume to identify its size and shape. The interfacial tension was calculated automatically by FAMAS analysis software, using the Young Laplace equation.

### 2.4. Formation of Oil-in-Water (O/W) Emulsions

Coarse emulsions were formed by using 1% (*w/w*) of LAEs in ultrapure water (95%, *w/w*, pH ≈ 7) as the aqueous phase and 5% (*w/w*) of the oil phase (soybean oil). The selected LAE was used to prepare the other set of emulsions with different concentrations of extracts (0.1–2%, *w/w*) or different oil concentrations (0.5–20%, *w/w*). To prevent microbial growth during storage, all emulsions were treated with 0.02% (*w/w*) sodium azide. The mixtures were then mixed by using a high-speed mixer at 10,000 rpm for 5 min (Polytron®, System PT 3100, Kinematica AG, Lucerne, Switzerland). To form the fine emulsions, coarse emulsions were passed through a single-stage high-pressure homogenizer (NanoVater, NV200, Yoshida Kikai, Nagoya, Japan) at 100 MPa for four passes, under standardized conditions, and stored at 5 °C until measurements.

### 2.5. Emulsions Characterization and Stability Evaluation
### 2.5.1. Characterization of Emulsions

The volume mean diameter, $d_{4,3}$ ($=\sum n_i\, d_i^4/\sum n_i\, d_i^3$), where $n_i$ refers to the number of droplets with diameter, $d_i$, and droplet size distributions of emulsions were determined by a static laser diffraction particle size analyzer (LS 13,320, Beckman Coulter, Brea, CA, USA). The droplet charge known as ζ-potential of each emulsion was measured by using a dynamic light-scattering particle size analyzer (Zetasizer, Nano ZS, Malvern Instruments

Ltd., Worcestershire, UK). Briefly, the emulsion was diluted (1:100) with ultrapure water to avoid multiple scattering effects. Then 1 mL of the diluted emulsion was placed in a disposable folded capillary cell (DTS 1070) and equilibrated at 25 °C for 60 s. The refractive index of the oil and aqueous phases was fixed at 1.432 and 1.330, respectively.

2.5.2. Evaluation of Emulsions Stability

The stability of the emulsion was evaluated by monitoring the volume mean droplet diameter ($d_{4,3}$) and the visual aspect of the emulsion for 30 days (Day 0, 7, 15, 21, and 30) at two different storage temperatures (5 and 25 °C).

*2.6. Statistical Analysis*

All the experiments were carried out in 3 replicates, and the mean ± standard deviation was indicated. One-way analysis of variance (ANOVA) with the Duncan Multiple Range test as reference was solely conducted to assess significant differences among variables at the 95% confidence level ($p \leq 0.05$) for the study of physicochemical characterization of the *Limnophila aromatica* extracts. For this purpose, SPSS statistic software version 28.0 (IBM Corp., Armonk, NY, USA) was used as the standard program.

**3. Results and Discussion**

*3.1. Physicochemical Properties of Limnophila Aromatica Extracts*

3.1.1. Extraction Yields and Chemical Compositions of Limnophila Aromatica Extracts

In this study, *Limnophila aromatica* extracts (LAEs) were obtained by solvent extraction, using different concentrations of aqueous ethanol, including 0, 25, 50, 75, and 99.5% ($v/v$). As shown in Table 1, the extraction yield ranged from 1.22% ($w/w$) for absolute ethanol to 9.53% ($w/w$) for 50% ($v/v$) aqueous ethanol. The yield of extraction by various concentration of aqueous ethanol increased in the following order: 99.5% < 0% < 75% < 25% < 50%. It was observed that extraction yields were raised (0% < 25% < 50%), while the polarity of the ethanol used in the extraction was augmented. However, the yield of extraction was also enhanced (99.5% < 75% < 50%) when increasing the water concentration. This may be caused by the increased solubility of proteins and carbohydrates in water compared to ethanol [41]. The use of water and ethanol in combination may facilitate the extraction of soluble compounds in water and/or organic solvent. These findings also agree with the previous studies on the extraction yield of defatted *Limnophila aromatica* [35] and some medicinal plants [42].

**Table 1.** Extraction yields (%, $w/w$), surface-active compounds (%, $w/w$), and total ash content (%, dry basis) of *Limnophila aromatica* extracts (LAEs) obtained by various aqueous ethanolic extractions. Average values in the same column with different letters are significantly different at a 95% confidence interval ($p \leq 0.05$).

| Ethanol Concentration | Extraction Yield | Protein Content | Saponin Content | Phenol Content | Ash Content |
|---|---|---|---|---|---|
| 0% ($v/v$) | 5.95 ± 0.55 [d] | 5.38 ± 0.27 [a] | 7.75 ± 0.20 [d] | 3.70 ± 0.04 [e] | 24.50 ± 1.88 [b] |
| 25% ($v/v$) | 8.67 ± 0.37 [b] | 4.97 ± 0.33 [ab] | 8.17 ± 0.10 [d] | 5.37 ± 0.08 [d] | 21.96 ± 1.96 [bc] |
| 50% ($v/v$) | 9.53 ± 0.15 [a] | 4.56 ± 0.06 [bc] | 17.17 ± 0.37 [c] | 9.20 ± 0.13 [c] | 19.69 ± 0.73 [c] |
| 75% ($v/v$) | 7.72 ± 0.09 [c] | 4.36 ± 0.06 [c] | 20.14 ± 0.40 [b] | 11.68 ± 0.12 [b] | 14.74 ± 0.55 [d] |
| 99.5% ($v/v$) | 1.22 ± 0.08 [e] | 3.81 ± 0.29 [d] | 23.87 ± 0.54 [a] | 12.79 ± 0.07 [a] | 30.86 ± 2.20 [a] |

Surface-active compounds such as saponin, protein, and phenolic are strongly related to the emulsifying properties of the extracts because their chemical structures contain both hydrophilic and hydrophobic moieties [4,43]. However, the residual elements were also strongly related to the emulsion instability [26]. Therefore, it is necessary to characterize both surface-active and residual substances in the extracts (Table 1).

As shown in Table 1, the extract obtained by the absolute ethanol contained the highest saponin (23.87 ± 0.54%, *w/w*), followed by 75, 50, 25, and 0% (*v/v*) aqueous ethanol, respectively. It indicated that the saponin contents significantly increased when reducing the polarity of solvent by increasing the ethanol concentration. The structure of saponins isolated from medicinal plants generally consists of different types of aglycone molecules (water-non-soluble components) and one or two sugar moieties (water-soluble components). In particular, the hydrophobic saponins are easily dissolved in polar solvents, while hydrophilic saponins are well soluble in nonpolar solvents [44]. Therefore, *Limnophila aromatica* plants might contain many more semi-polar and polar saponins. Moreover, this study corresponds with Do et al., who found that more polar solvents were better for saponin extraction from *Codonopsis javanica* root than less polar solvents [45].

The highest total phenol content of the extract was obtained from absolute ethanol (12.79 ± 0.07%, *w/w*), followed by 75, 50, 25, and 0% (*v/v*) aqueous ethanol, respectively (Table 1). This trend showed that the total phenol content of LAE was increased while decreasing the water concentration during extraction. It may indicate the presence of more non-phenol compounds, such as carbohydrate and terpene, in water extract than in other extracts. It may be attributable to the complex formation of certain phenolic compounds in the extracts that are soluble in ethanol. It is also well-known that using a significantly greater percentage of ethanol enhances polyphenols extraction yield [35,46].

The highest protein contents (5.38 ± 0.27 and 4.97 ± 0.33%, *w/w*) were obtained by water extraction and 25% (*v/v*) aqueous ethanolic extraction, respectively, while the lowest (3.81 ± 0.29%, *w/w*) was acquired from absolute ethanol. The results showed that the protein content was decreased while increasing the polarity of the solvent, indicating the precipitation of protein by absolute ethanol. Similarly, the protein was reported as unstable and less soluble in polar solvents such as absolute ethanol [47].

The ash content of *Limnophila aromatica* extracts referred to the quantity of inorganic material present, both internally by plants and during the extraction procedures. Among all LAEs, the absolute ethanol extract contained the highest ash content (30.86 ± 2.20%, dry basis), while the extract obtained by 75% aqueous ethanolic extraction had the lowest ash content (14.74 ± 0.55%, dry basis). Ash is the inorganic residue that remains after removing the water and organic materials by heating and is used to estimate the total mineral contents [48]. Nevertheless, the high ash contents represent the impurity substances that significantly affect the forming and stabilizing of emulsion [26].

### 3.1.2. Interfacial Activities of Limnophila Aromatica Extracts

The existence of surface-active compounds that may rapidly adsorb at the oil/water interface was therefore exclusively attributable to the interfacial activities of *Limnophila aromatica* extracts. Saponins are tiny surfactants whose structure consists of hydrophilic materials linked to hydrophobic materials. As a result, they may adsorb effectively at the oil/water interface, lowering the interfacial tension between the oil and water phases [6]. Phenolic compounds are tiny molecular substances that displace quickly at the oil/water interface. They generally have less surface activity than saponins, so they are rarely used as main emulsifiers [49]. In contrast, proteins have a greater particle size and take longer to adsorb at the oil/water interface. Due to their larger adsorption energy, proteins may more effectively attach to the droplet interface, improving emulsion stability [21]. Owing to the complex constitution of *Limnophila aromatica* extracts, it was challenging to identify the specific component responsible for interfacial tension reductions in this investigation.

As shown in Figure 1a, all *Limnophila aromatica* extracts (LAEs) could reduce the interfacial tension at the soybean/water interface. The interfacial tensions of LAEs were closed to ultrapure water when using a low concentration (≤0.01%, *w/w*) of the extracts, reflecting poor adsorption of surface-active compounds at the oil/water interface. At the medium concentration (1%, *w/w*), the lowest interfacial tension (12.58 ± 0.22 mN/m) was obtained by LAE-99.5, while the highest interfacial tensions were obtained (16.18 ± 0.06 and 15.86 ± 0.22 mN/m) by LAE-0 and LAE-25, respectively (Figure 1b). As expected, the

highest saponin and phenol content acquired from absolute ethanol showed significant effects on the interfacial properties of the extract. In contrast, the highest protein contents of the extracts obtained by water and 25% (*v/v*) aqueous ethanol did not significantly affect the interfacial activities of the extracts. However, LAE-50 could also reduce its interfacial tension to 13.29 ± 0.20 mN/m, even though it did not contain the highest saponin, protein, or phenol contents. At the highest concentration of 3% (*w/w*), the interfacial tension was decreased to 10.12–13.2 mN/m. This result indicated that the interfacial tension values of all LAEs are in the same range of minimal interfacial tension (7–16.3 mN/m) of other effective natural emulsifiers extracted from the plants and agro-industrial by-products, including red beet, oat bran, argan press cake, sugar beet, argan shell, olive oil cake, liquorice root, and bagasse [20–26,50].

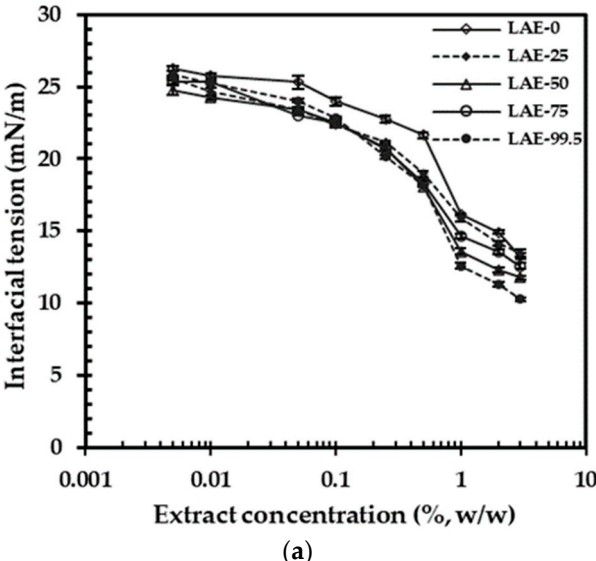 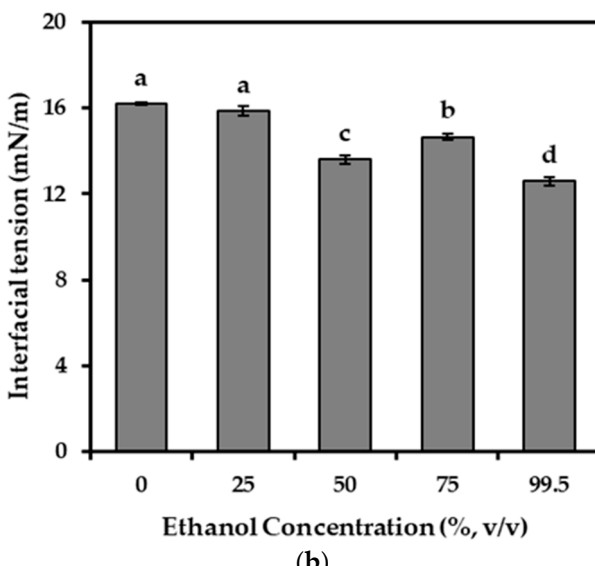

(**a**)　　　　　　　　　　　　　　　　　　(**b**)

**Figure 1.** Interfacial tension at soybean oil/water interfaces as a function of *Limnophila aromatica* extract concentrations: (**a**) various concentrations (0.005–3%, *w/w*) and (**b**) 1% (*w/w*) of *Limnophila aromatica* extracts. LAE refers to *Limnophila aromatica* extract and 0, 25, 50, 75, and 99.5 mean ethanol concentration (%, *v/v*). The interfacial tension between soybean oil and ultrapure water was approximately 26 mN/m. The different letter indicates a significantly different ($p \leq 0.05$).

### 3.2. Emulsifying Properties of Limnophila Aromatica Extracts (LAEs)

It was impossible to choose an effective *Limnophila aromatica* extract for preparing emulsions based on their chemical compositions and interfacial activities. Therefore, all LAEs (1%, *w/w*) were used for preparing oil-in-water (O/W) emulsions with 5% (*w/w*) soybean oil. The volume mean droplet diameter ($d_{4,3}$) of the emulsions was measured immediately (Day 0) and after one week (Day 7) of storage at 5 °C to compare the emulsifying properties of the extracts. The electrical charge ($\zeta$-potential) measurements of fresh emulsions were also conducted.

All LAEs were successfully used as emulsifiers to prepare emulsions ($d_{4,3} < 1$ μm) with the volume mean diameter ($d_{4,3}$) ranging from 273 to 747 nm, except for emulsions using LAE-99.5 (absolute ethanol), which showed the highest droplet diameter ($d_{4,3} = 3269 \pm 29$ nm) among the samples. LAE-50 was the most effective emulsifier, as it could reduce the volume mean diameter ($d_{4,3}$) of fresh emulsion to the minimum values of approximately 273 nm, followed by LAE-75, LAE-25, LAE-0, and LAE-99.5, respectively (Figure 2a). Because LAE-99.5 did not successfully formulate the submicron emulsions, the stability of emulsions prepared by this extract was not observed. Surprisingly, the droplet size ($d_{4,3}$) of emulsions obtained by LAE-0, LAE-25, or LAE-50 were significantly increased after 7 days of storage at 5 °C, while LAE-75 was able to stabilize the emulsions without significant change of droplet size at the same storage condition and times (Figure 2b).

Independent of the type of extract utilized to stabilize emulsions, bimodal droplet size distributions were observed, indicating that emulsions might contain varying droplet sizes (Figure 3a). Nevertheless, there was almost no change in the droplet size distribution of emulsions using LAE-75 between Day 0 and Day 7, leading to maintained stability at the emulsion interface and coalescence inhibition. These results were supported by the visual appearance of the emulsions, which showed significant flocculation and coalescence after 7 days of storage at 5 °C (Figure 3b). A creaming layer popped up on the top of emulsions using LAE-0, LAE-25, or LAE-50, and the serum of the aqueous phase also could be seen. However, emulsions using LAE-75 as an emulsifier remained nearly unchanged in regard to their visual appearance upon storage. Therefore, LAE-75 was selected as the emulsifier to be utilized for further experiments.

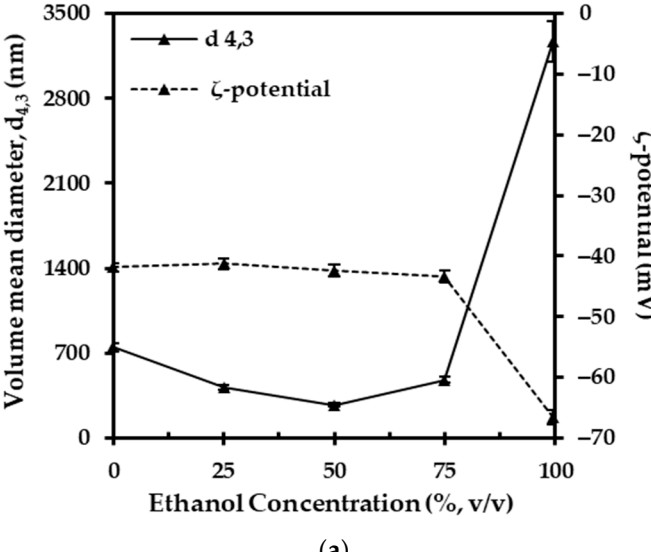
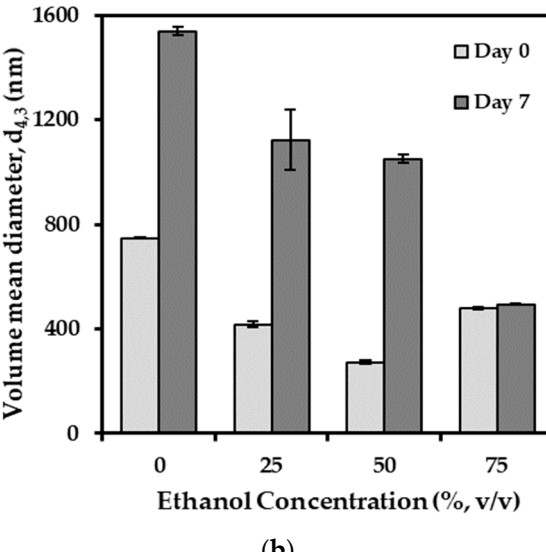

**Figure 2.** Effect of ethanol concentration during extraction of *Limnophila aromatica* on (**a**) volume mean diameter ($d_{4,3}$) and $\zeta$-potential of emulsions measuring immediately after homogenization and (**b**) volume mean diameter ($d_{4,3}$) of newly prepared emulsions (Day 0) and after 7 days of storage at 5 °C.

The interfacial layer properties of oil droplets and the continuous phase composition significantly impact the physical stability of emulsions [51]. Rapid screening of charges at the oil/water interfaces, induced by increasing salt concentration in electrostatically stabilized emulsion, generally results in flocculation and coalescence of emulsions [21,22]. Furthermore, flocculation is usually caused by depleting non-adsorbing emulsifiers, leading to emulsion destabilization [52]. In this study, the emulsions prepared by extracts containing a high concentration of inorganic substances indicated less emulsion stability, as confirmed by the total ash content determination (Table 1). Therefore, we believe that the residual ionic composition of *Limnophila aromatica* extracts (LAEs) causes the electrostatic screening of droplets' interface to influence emulsions' stability significantly.

The electrical charge of emulsions effectively influences their stability under various storage conditions. The high negative charge mainly indicates that the emulsifier layer produces more repulsive forces between emulsion droplets, inhibiting their coalescence and leading to stabilizing the emulsion [53]. A study by Losso et al. indicated that the emulsions had good stability when their $\zeta$-potential values ranged between −41 and −50 mV [54]. In contrast, the $\zeta$-potential of LAEs emulsions did not respond to their stability characteristics. All emulsions using LAEs consisted of a similarity $\zeta$-potential value (−41 to −43 mV), except for LAE-99.5, which showed the highest negative charge (−67 mV) of emulsions (Figure 2a). The negative charge of emulsions might be induced by surface-active substances such as protein, saponin, and phenolic compounds of the extracts. Protein composition generally consists of acidic and basic groups, leading to a high negative charge (i.e., the

ζ-potential of soy and chickpea proteins is approximately −40 mV) [55,56]. Typically, ζ-potential of emulsions prepared by saponin, such as quillaja, were highly negative because their structures consisted of the carboxylic group [57]. However, the electrical charge of saponin emulsions is also affected by other compounds with anionic residues in the crude extracts. Böttcher et al. related that reducing the negative charge of emulsion from −70 to −50 mV at pH 7 was obtained when they purified the extracts by removing anionic non-saponin elements [58].

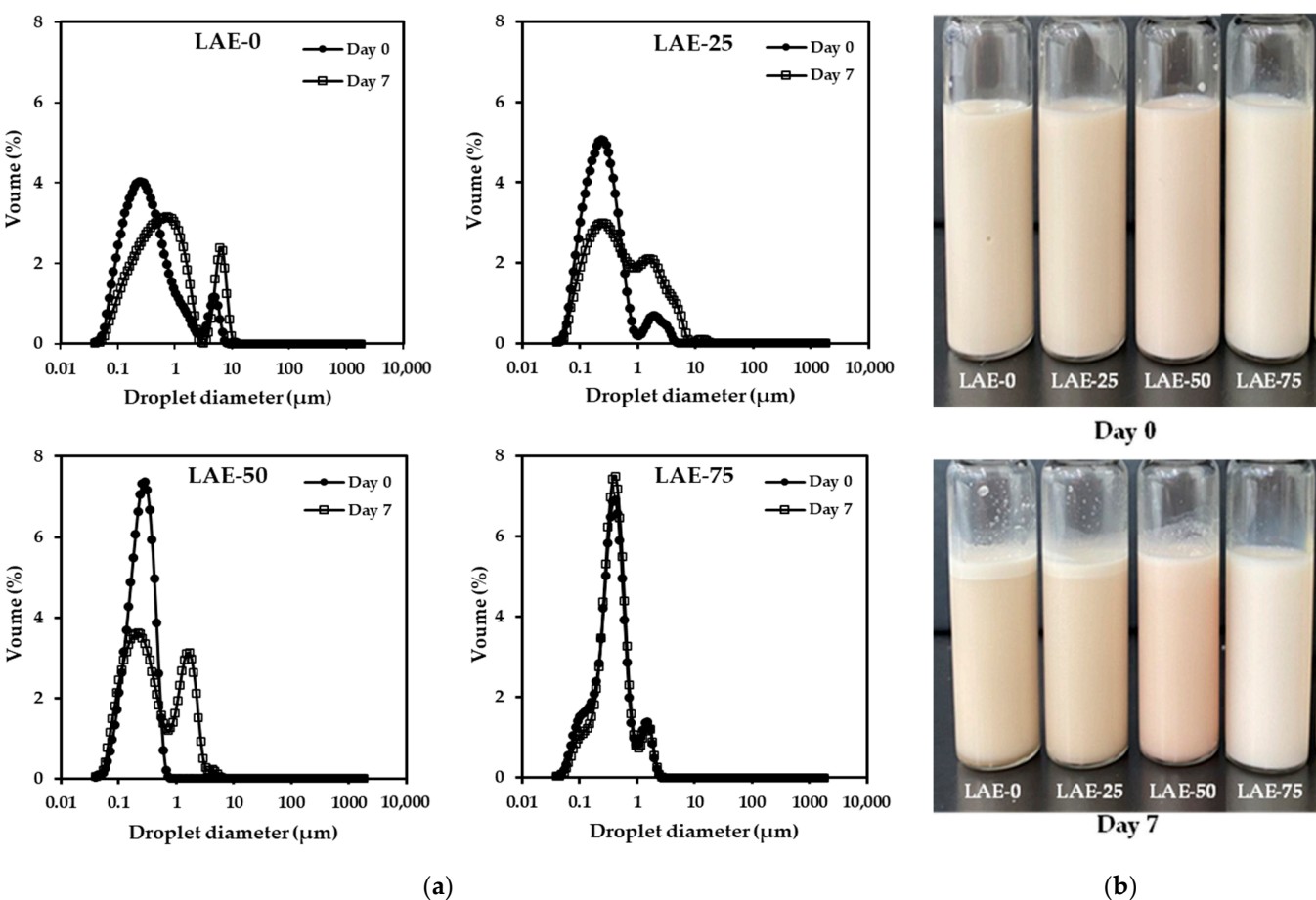

**(a)**                                                                                   **(b)**

**Figure 3.** (**a**) Droplet size distribution and (**b**) visual appearance of emulsions prepared by using 1% (*w/w*) LAEs and 5% (*w/w*) soybean oil immediately after homogenization and after Day 7 of storage at 5 °C. LAE refer to *Limnophila aromatica* extract and 0, 25, 50, 75, and 99.5 indicate ethanol concentration (%, *v/v*).

### 3.3. Effect of Extract Concentration on Emulsion Formation and Stabilization

According to the results described in the above section, *Limnophila aromatica* extract obtained by 75% (*v/v*) aqueous ethanolic extraction (LAE-75) was selected as the emulsifier to prepare emulsions with various concentrations of extracts (0.1–2%, *w/w*) and 5% (*w/w*) soybean oil. Figure 4a shows that the volume mean diameter ($d_{4,3}$) of emulsions was decreased from $836 \pm 19$ nm to reach the minimum value ($424 \pm 5$ nm) when increasing the LAE-75 concentration from 0.1 to 1% (*w/w*). By further increasing the concentration of LAE-75 to 1.5 and 2% (*w/w*), the droplet size ($d_{4,3}$) of emulsions was increased to $659 \pm 83$ nm and $718 \pm 72$ nm, respectively. Interestingly, high concentrations of extract, such as 1.5 and 2% (*w/w*), could not stabilize the emulsions, while 1% (*w/w*) was able to stabilize emulsions for only 7 days of storage at 5 °C. The emulsions using the low concentrations of extract (≤0.5%, *w/w*) remained at pretty much the same droplet size ($d_{4,3}$) after storage for 15 days at 5 °C (Table 2). These results were confirmed with the

visual appearance of emulsions, as shown in Figure 4b. Oiling-off and creaming were seen from emulsions using 1.5 and 2% (*w/w*) LAE-75 at Day 7 and 1% (*w/w*) at Day 15, while emulsions using lower concentrations did not change appearance after 15 days.

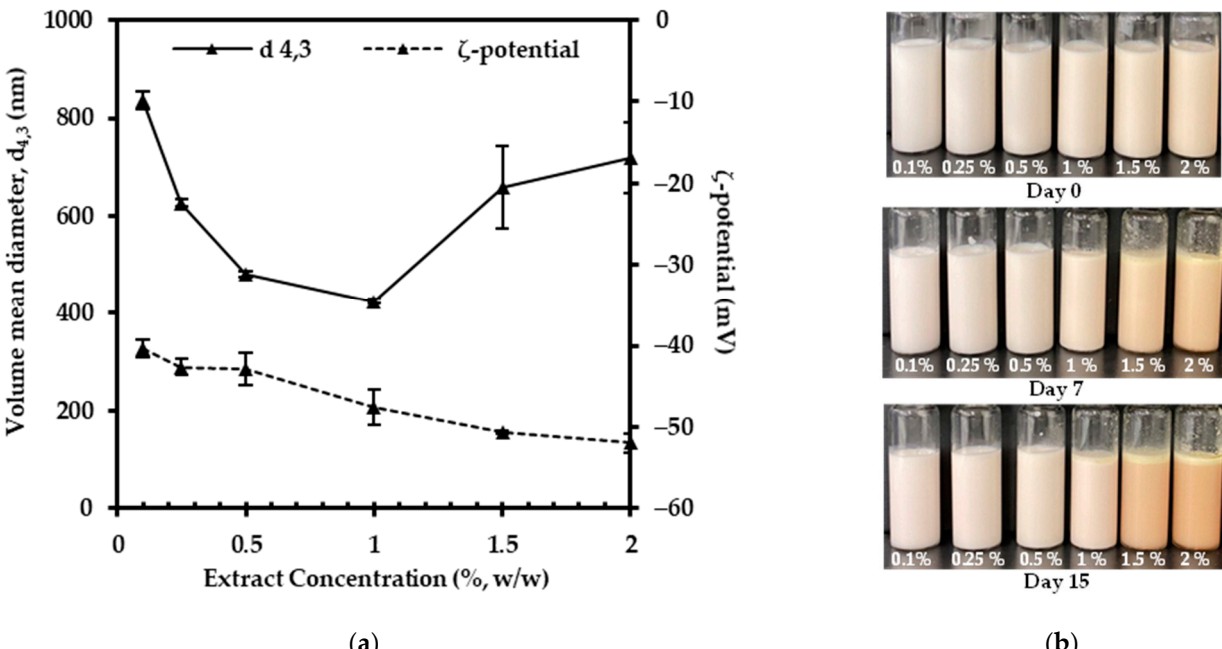

|            (**a**)            |            (**b**)            |

**Figure 4.** Effect of 75% (*v/v*) aqueous ethanolic *Limnophila aromatica* extract (LAE-75) concentration on (**a**) volume mean diameter ($d_{4,3}$) and $\zeta$-potential of emulsions measured immediately after homogenization; and (**b**) visual appearance of emulsions at Day 0, 7, and 15 of storage at 5 °C.

**Table 2.** Volume mean diameter, $d_{4,3}$ (nm) of emulsions stabilized by different concentration of 75% (*v/v*) aqueous ethanolic *Limnophila aromatica* extract (LAE-75) with 5% (*w/w*) soybean oil at Day 0, 7, and 15 of storage at 5 °C.

| Extract Concentration | Day 0 | Day 7 | Day 15 |
|---|---|---|---|
| 0.1% (*w/w*) | $836 \pm 19$ | $834 \pm 5$ | $832 \pm 16$ |
| 0.25% (*w/w*) | $627 \pm 8$ | $640 \pm 12$ | $638 \pm 5$ |
| 0.5% (*w/w*) | $481 \pm 6$ | $486 \pm 6$ | $483 \pm 3$ |
| 1% (*w/w*) | $424 \pm 5$ | $509 \pm 5$ | * |
| 1.5% (*w/w*) | $659 \pm 83$ | * | * |
| 2% (*w/w*) | $718 \pm 72$ | * | * |

* The droplet size of emulsion was not analyzed owing to the oiling-off or creaming observation.

As a result, the droplet size of emulsions was large when using a low concentration of extract (<0.5% *w/w*) as the emulsifier, because the surface-active compounds to cover the oil interfaces were insufficient. However, using a high concentration of extract (>0.5%, *w/w*) led to flocculation and coalescence upon storage, due to the increasing of un-adsorbed surface-active substances and destabilizing agents (i.e., mineral) in the emulsions [25,59].

The electrical charge of emulsions prepared by various concentrations of LAE-75 ranged from −40 to −52 mV for 0.1 and 2% (*w/w*), respectively (Figure 4a). The high negative charge of emulsions remarkably improves emulsion stability by preventing coalescence and flocculation [53]. In contrast, the emulsions using 2% (*w/w*) LAE-75 indicated the highest negative charge with less stability. As discussed in the previous sections, the negative charge of emulsions might be induced by both the charge of surface-active substances and anionic residual substances [52]. As reported in Table 1, the total ash content of LAE-75 was $14.74 \pm 0.55\%$ (dry basis), meaning that the extract might contain a high amount of anionic residual compounds.

### 3.4. Effect of Oil Concentrations on Emulsion Formation and Stabilization

The oil-in-water (O/W) emulsions were prepared by using 0.5% (*w/w*) LAE-75 as an emulsifier with different concentrations of soybean oil (2.5 to 20%, *w/w*). The droplet size ($d_{4,3}$) of the emulsion was increased, while increasing the oil concentration (Figure 5a), and ranged between 250 and 2349 nm from 2.5 and 20% (*w/w*), respectively. The emulsions using 20% (*w/w*) oil showed oiling-off a few hours after homogenization, while the 10% (*w/w*) oil could stabilize the emulsions for 3 days and showed oiling-off at Day 7 (Figure 5b). However, LAE-75 (0.5%, *w/w*) could stabilize 2.5 and 5% (*w/w*) soybean oil up to 7 days, with a droplet size ($d_{4,3}$) of approximately 250 and 450 nm, respectively. The instability of emulsions using high oil concentration is typically caused by the extract's insufficient concentration and the increasing emulsion viscosity, leading to disruption efficiency during homogenization [60]. Therefore, LAE-75 was able to stabilize oil-in-water (O/W) emulsions with a concentration of oil up to 5% (*w/w*). We suggest that the stability of emulsion using a high oil concentration could strengthen by increasing the concentration of purified 75% (*v/v*) aqueous ethanolic *Limnophila aromatica* extract (LAE-75).

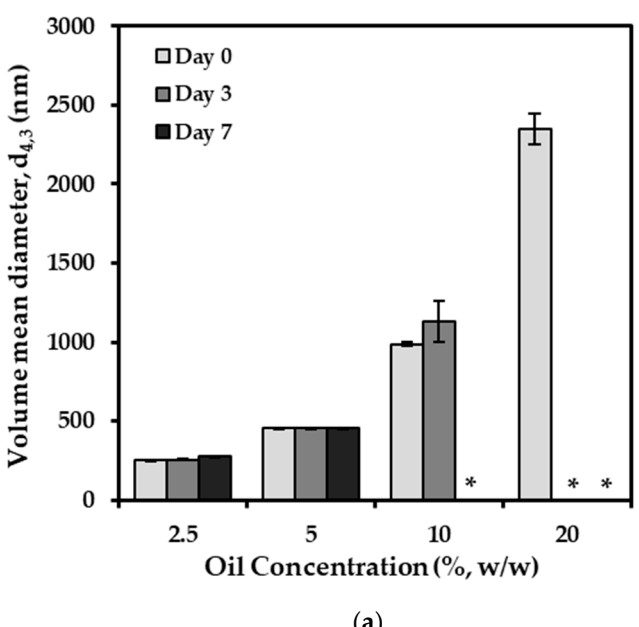

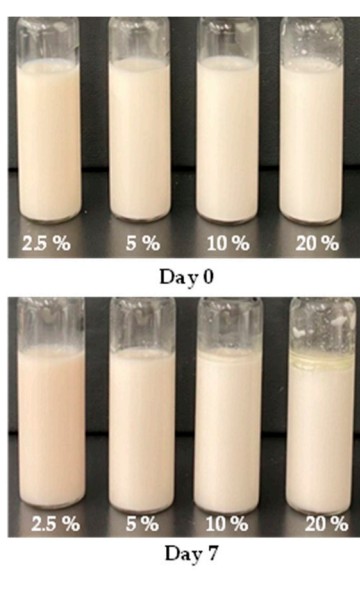

(**a**)                                                                 (**b**)

**Figure 5.** Effect of oil concentration on (**a**) volume mean diameter ($d_{4,3}$) and (**b**) visual appearance of emulsions using 0.5% (*w/w*) LAE-75 at Day 0, 7, and 15 of storage at 5 °C. * The droplet size of emulsion was not analyzed, owing to the oiling-off or creaming observation.

### 3.5. Stability of Oil-in-Water (O/W) Emulsion

The droplet size ($d_{4,3}$) of oil-in-water (O/W) emulsions prepared by 0.5% (*w/w*) of 75% (*v/v*) aqueous ethanolic *Limnophila aromatica* extract (LAE-75) with 5% (*w/w*) soybean oil was continuously measured for 30 days of the storage at 5 and 25 °C. Table 3 shows that the droplet size was completely unchanged upon the storage at either 5 or 25 °C. Therefore, LAE-75 is potentially used as a natural emulsifier to formulate and stabilize oil-in-water (O/W) emulsions.

**Table 3.** Volume mean droplet diameter, $d_{4,3}$ (nm), of oil-in-water (O/W) emulsions stabilized by 0.5% (*w/w*) LAE-75 and 5% (*w/w*) soybean stored at different temperatures for 30 days.

|  | Day 0 | Day 7 | Day 15 | Day 21 | Day 30 |
|---|---|---|---|---|---|
| 5 °C | 435 ± 1 | 446 ± 5 | 460 ± 4 | 481 ± 2 | 482 ± 18 |
| 25 °C | 442 ± 5 | 466 ± 1 | 484 ± 7 | 482 ± 3 | 496 ± 6 |

## 4. Conclusions

In summary, this study is the first report using *Limnophila aromatica* extracts (LAEs) as the natural emulsifiers to stabilize the oil-in-water (O/W) emulsions. Except for the extract using absolute ethanol (LAE-99.5), all LAEs were able to produce emulsions with a highly negative charge. However, only 75% (*v/v*) aqueous ethanolic *Limnophila aromatica* extract (LAE-75) could stabilize emulsions for up to 7 days at 5 °C. We noticed that LAE-75 did not contain the highest concentration of surface-active compounds, but it consisted of the lowest ash content. These results indicated that the emulsifying properties of LAE did not depend only on surface-active compounds and interfacial activities. Residual substances in the extracts could be the destabilized agents that caused the instability of emulsions. The optimum concentration of LAE-75 for stabilizing emulsions was 0.5% (*w/w*). By using this concentration, LAE-75 could stabilize emulsions up to 5% (*w/w*) soybean oil for 30 days against two different storage temperatures (5 and 25 °C) without significantly changing droplet size (435–496 nm) and visual appearance. In conclusion, *Limnophila aromatica* extract has a potential application as a new source of natural emulsifier. For future work, we suggest using purified *Limnophila aromatica* extracts as the emulsifiers by removing inorganic residuals to improve the emulsifying abilities of the extracts to stabilize nanoemulsions (droplet size less than 200 nm) with high oil concentrations.

**Author Contributions:** Conceptualization, R.S., N.T. and M.B.; methodology, R.S., L.d.O.F. and M.B.; software, R.S.; validation, M.N. and M.A.N.; formal analysis, R.S.; investigation, R.S.; resources, M.A.N.; data curation, R.S.; writing—original draft preparation, R.S.; writing—review and editing, L.d.O.F. and N.T.; visualization, R.S.; supervision, M.N. and M.A.N.; funding acquisition, M.A.N. All authors have read and agreed to the published version of the manuscript.

**Funding:** This research received no external funding.

**Institutional Review Board Statement:** Not applicable.

**Informed Consent Statement:** Not applicable.

**Data Availability Statement:** Not applicable.

**Acknowledgments:** Rasmey Soeung acknowledges the Ministry of Education, Culture, Sports, Science and Technology (MEXT) of Japan for providing the scholarship to conduct the current research under the MEXT Special Scholarship Program on Trans-world Professional Human Resources Development Program on Food Security & Natural Resources Management (TPHRD) for a Doctoral Course.

**Conflicts of Interest:** The authors declare no conflict of interest.

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
