# Peer review of "Limnophila aromatica Crude Extracts as Natural Emulsifiers for Formation and Stabilizing of Oil-in-Water (O/W) Emulsions"

_colloids, doi:10.3390/colloids6020026_

Round 1
Reviewer 1 Report
The manuscript describes the preparation and the colloidal stability of oil-in-water emulsions stabilized by plant extracts. The text is mostly clear and concise and may be of interest to the audience working on natural emulsifiers. Hence, I support its publication after some revisions, as stated below:
- Considering that the authors report a phase separation just after emulsion preparation for LAE-99.5 sample, the consistency for the reported zeta potential value for this sample (Figure 2a) should be checked. Is this a realistic number? Is the sample really emulsified during the measurement or is it a phase-separated sample?
- Considering that, basically, oil droplets in an emulsion can be kinetically stabilized by electrostatic and/or steric effects, with the later being discarded at this particular case, what could explain emulsions, prepared by the same procedure, with similar zeta potential values and different long-term stability (Figure 2a and b)?
- On page 9, line 374, I think the more appropriate term would be "creaming" rather than "creamy". Also, the term "size of emulsions", which appeared in line 377 of the same page, should be avoided, since the size counts for the oil droplets and not for the whole system itself.
- In Table 2, page 10, I would avoid using "non detected" for samples where emulsification was not properly achieved. Since it is a result based on light scattering, some readers can understand this as being something related with size ranges detected by the technique, i.e. droplets are formed but are too small or too big to be measured. Same for Figure 5.
- The ash content in the different extracts seems not to influence the zeta potential values of the emulsions, although authors claim in the conclusion that it may be important for long-term stability. What kind of microstructural or molecular changes could be caused by the presence of ash?
Author Response
RESPONSES TO REVIEWER 1
First of all, the authors would like to express our deepest gratitude to the reviewer for taking the time to thoroughly review and appraise your comments and suggestion to improve the quality of our manuscript. Please kindly see below, for a point-by-point response to your comments and questions, where the page and line numbers mentioned hereafter refer to the number in the revised (clear version) of the manuscript (pdf file (these are the final page and line numbers)).
Comments and Suggestions for Authors
The manuscript describes the preparation and the colloidal stability of oil-in-water emulsions stabilized by plant extracts. The text is mostly clear and concise and may be of interest to the audience working on natural emulsifiers. Hence, I support its publication after some revisions, as stated below:
Response: The authors deeply appreciate your kindness in spending your valuable time reviewing this manuscript. We really appreciate your interest in our research and your encouragement.
Considering that the authors report a phase separation just after emulsion preparation for LAE-99.5 sample, the consistency for the reported zeta potential value for this sample (Figure 2a) should be checked. Is this a realistic number? Is the sample really emulsified during the measurement or is it a phase-separated sample?
LAE-0 LAE-25 LAE-50 LAE-75 LAE-99.5 |
Response: The authors acknowledge the reviewer’s concern in raising this point. The value reported for the zeta-potential (c.a. -67 mV) of emulsions prepared from LAE-99.5, as shown in Fig. 2 (a), is confirmed as a realistic number. It was measured three times for every replicate, where the value determined for every replicate was like the average value with standard deviation reported on the graph. The sample was completely emulsified after its formulation regarding the visual aspect of the LAE-99.5 emulsion. Thus, by naked eyes observation, the whole LAE-99.5 emulsion was completely homogeneous, without signals of phase separation or oiling-off when the measurements of these samples were conducted immediately after preparation. A minor oiling off was noticed several hours later. As follow, the picture of freshly prepared emulsions is shown to confirm the homogeneity of the samples where the intact aspect of the LAE-99.5 is seen. The authors would like to apologize for the mistake written on this point in our previous manuscript. We have justified the sentence in the revised manuscript to be more understandable (Page 8, Line 324-327).
Considering that, basically, oil droplets in an emulsion can be kinetically stabilized by electrostatic and/or steric effects, with the later being discarded at this particular case, what could explain emulsions, prepared by the same procedure, with similar zeta potential values and different long-term stability (Figure 2a and b)?
Response: Thanks in advance for this interesting question. As a crude, non-purified extract, a series of impurities are frequently present, consisting of a heterogeneous blending that might be compounded by minerals, fats, phenols, proteins, sugars, tannins, etc. This difference in the crude extract composition was also clearly shown for Limnophila aromatica as the present subject of our paper, where saponins, phenolic compounds, protein, and ash contents differed among the crude extracts. Thus, despite the similarity found for the zeta-potential values between the extracts, their whole composition was different, which can directly affect the stability of the emulsions along with time, as was demonstrated here. Owing to this, considering the results discussed in our paper, we hypothesize that the physical stability of the emulsions stabilized by crude emulsifiers is heavily affected by their overall composition and unknown impurities. Thus, it is difficult to judge the long-term stability just by the negative charge value of the crude extracts. As a non-purified and heterogeneous extract, the physical stability might affect by different factors than solely by the electrical charge, as easily happen for purified emulsifiers.
On page 9, line 374, I think the more appropriate term would be "creaming" rather than "creamy". Also, the term "size of emulsions", which appeared in line 377 of the same page, should be avoided, since the size counts for the oil droplets and not for the whole system itself.
Response: We are so grateful to the reviewer for pointing out these issues. Thereby, following your advice, we replaced all 'creamy' terminology with 'creaming' in all the cases (Page 9, Line 341&387). Similarly, we changed the terminology 'size of emulsions' to 'droplet size of emulsions' to avoid misunderstanding by the readers (Page 10, Line 390).
In Table 2, page 10, I would avoid using "non detected" for samples where emulsification was not properly achieved. Since it is a result based on light scattering, some readers can understand this as being something related with size ranges detected by the technique, i.e. droplets are formed but are too small or too big to be measured. Same for Figure 5.
Response: Thank you for your valuable comments in order to improve the understanding of our work. Considering this, we replaced all the "ND" abbreviations with the asterisk symbol (*). Thus, at the bottom of the table or graphic, a footnote was indicated by the asterisk symbol (*), indicating that "the droplet size of emulsion was not analyzed owing to the oiling-off or creaming observation" (Page 10, Line 411 & Page 11 Line 429-430).
The ash content in the different extracts seems not to influence the zeta potential values of the emulsions, although authors claim in the conclusion that it may be important for long-term stability. What kind of microstructural or molecular changes could be caused by the presence of ash?
Response: Thank you very much for raising these important points. Generally, the higher zeta potential values mean that the emulsifier layer yields more repulsive force between emulsion droplets, thus preventing their coalescence, while a neutral zeta potential means that the emulsifier covered interfaces are more likely to destabilize, resulting in the formation of larger particles. Saponins, proteins, and phenolic compounds are typically responsible for the negative charges as they tend to be negatively charged at pH 7. However, the zeta potential values of all emulsions prepared from Limnophila aromatica extracts ranged from -41 to -67 mV, independently of their stability. The emulsions prepared by extracts containing high ash content indicated less stability. Ash is the inorganic residue that remains after removing the water and organic materials by heating and is used to estimate the total mineral contents (Pomeranz, Y.; Meloan, C. E. In Food Analysis, 1994). Most minerals are made up of a cation (a positively charged ion) or several cations and an anion (a negatively charged ion) or an anion complex. Furthermore, increasing salt (one of the mineral substances) concentration in electrostatically stabilized emulsions generally induces rapid screening of charges at the oil/water interfaces, resulting in flocculation and coalescence of emulsions (Ralla, T.; Salminen, H.; Edelmann, M.; Dawid, C.; Hofmann, T.; Weiss, J. Food Hydrocoll., 2018;). Moreover, when we increased the concentration of the LAE-75 for emulsion formation, a higher negative charge was seen which indicated that the LAE-75 might contain more anionic residual. A study by Böttcher, S.; Keppler, J. K.; Drusch, S. Colloids Surf. A Physicochem. Eng. Asp., 2017, described that reducing the negative charge of emulsion from -70 mV to -50 mV at pH 7 was obtained when they purified the extracts by removing anionic non-saponin substances. Therefore, the ash contents could influence the zeta potential value as well important for the long-term stability of the emulsion.

Reviewer 2 Report
Manuscript ID: colloids-1662648
In the article entitled: “Limnophila aromatica Crude extracts as natural emulsifiers for formation and stabilizing of oil-in-water (O/W) emulsions” was examined the emulsifying performance of Limnophila aromatica crude extracts obtained using different ethanolic aqueous solutions.
This article is interesting, has been written in a compact manner. From the methodological point of view, the employed measurement techniques are appropriate to the adopted objective of the research work. The results obtained may have practical application.
Title
The title and the aim of the study are clearly constructed.
Abstract
The abstract includes the aim of the study, methods used in the experiment and contain the principal results and conclusions.
Introduction
The introduction describes the matter of the experiment accurately and clearly states the problem being investigated.
Methods
The data is well collected. The methods are clearly described, in the way which permits the research to be replicated. The sampling is appropriate and adequately described.
Statistical Analysis
All the experiments were carried in 3 replicates and one-way analysis of variance (ANOVA) with Duncan test. Has the absence of a series effect been demonstrated prior to the use of one-way analysis of variance?
Results and discussion
Generally, the results were discussed in a clear and legible way.
Conclusion
The authors correctly indicate, how the results are related to the studies.
References
The references are accurate.
Language
The article is correctly written.
In my opinion, the study deserves to be published in the Colloids and Interfaces.
Author Response
RESPONSES TO REVIEWER 2
First, the authors would like to express sincere thanks to the reviewer for kindly spending your valuable time on a thorough review of our manuscript. We believe that your comments and suggestion would lead to the improvement of our paper. Please kindly see below, for a point-by-point response to your comments and questions, where the page and line numbers mentioned hereafter refer to the number in the revised (clear version) of the manuscript (pdf file (these are the final page and line numbers)).
Comments and Suggestions for Authors
Manuscript ID: colloids-1662648
In the article entitled: “Limnophila aromatica Crude extracts as natural emulsifiers for formation and stabilizing of oil-in-water (O/W) emulsions” was examined the emulsifying performance of Limnophila aromatica crude extracts obtained using different ethanolic aqueous solutions.
This article is interesting, has been written in a compact manner. From the methodological point of view, the employed measurement techniques are appropriate to the adopted objective of the research work. The results obtained may have practical application.
Response: The authors sincerely applaud the reviewer for taking the time to review this manuscript. We really appreciate your interest in our research and your encouragement.
Title
The title and the aim of the study are clearly constructed.
Abstract
The abstract includes the aim of the study, methods used in the experiment and contain the principal results and conclusions.
Introduction
The introduction describes the matter of the experiment accurately and clearly states the problem being investigated.
Methods
The data is well collected. The methods are clearly described, in the way which permits the research to be replicated. The sampling is appropriate and adequately described.
Response: Thanks in advance for all your positive comments regarding our title, abstract, introduction, and methods.
Statistical Analysis
All the experiments were carried in 3 replicates and one-way analysis of variance (ANOVA) with Duncan test. Has the absence of a series effect been demonstrated prior to the use of one-way analysis of variance?
Response: The authors are thankful to the reviewer for raising this question. All the experiments were conducted in triplicate. However, one-way analysis of variance (ANOVA) having the Duncan Multiple Range test as reference for average comparison was solely used for the study of physicochemical characterization of the Limnophila aromatica extract. Statistical analysis was not applied to the emulsions' formation and stability studies. Regarding this matter, further clarification was done in the revised manuscript, as can be found on Page 5, Line 208-213.
Results and discussion
Generally, the results were discussed in a clear and legible way.
Conclusion
The authors correctly indicate, how the results are related to the studies.
References
The references are accurate.
Language
The article is correctly written.
Response: We are so grateful for all your encouraging feedback on our results and discussion, conclusion, references as well as language.
In my opinion, the study deserves to be published in the Colloids and Interfaces.
Response: The authors deeply appreciate your kindness and encouragement.

Reviewer 3 Report
The manuscript describe the use of a new ethanolic extract from Limnophila aromatica as emulsifier for o/w emulsions. The manuscript is readable and of interested for the reader. The rationale and experimental work is good.
At which temperature interfacial temperature was measured?
How many replicates for each emulsion formulation were prepared?
Were samples analysed for stability studies also protected by light or not?
Line 310 The oiling-off phenomenon is not strictly related to the size of the emulsion in the micrometric range for LAE-99.5.
It would be also relevant to discuss the effect of protein and phenolic compounds in relation to saponin content in determining the decrease in interfacial tension and emulsion formation and stability. This is also important since in the conclusion it is reported that: " the emulsifying properties of LAE did not depend only on surface-active compounds and interfacial activities".
Author Response
RESPONSES TO REVIEWER 3
First and foremost, the authors would like to express our sincere gratitude to the reviewer for taking the time to review our manuscript carefully and providing important comments, which have improved the overall quality of our text. Please kindly see below, for a point-by-point response to your comments and questions, where the page and line numbers mentioned hereafter refer to the number in the revised (clear version) of the manuscript (pdf file (these are the final page and line numbers)).
Comments and Suggestions for Authors
The manuscript describe the use of a new ethanolic extract from Limnophila aromatica as emulsifier for o/w emulsions. The manuscript is readable and of interested for the reader. The rationale and experimental work is good.
Response: The authors are so grateful to the reviewer for taking the valuable time to review our manuscript. We really appreciate your interest in our research and your encouragement
At which temperature interfacial temperature was measured?
Response: Thanks in advance for raising this question. The interfacial tension measurements were conducted at 25 °C. This additional information was then complemented in the “Materials and Methods” section of the present manuscript as can be found on Page 4, Line 174.
How many replicates for each emulsion formulation were prepared?
Response: Thanks for your question. Each emulsion was formulated in triplicate (i.e. three replicates).
Were samples analysed for stability studies also protected by light or not?
Response: The authors appreciate the reviewer’s question. The emulsions samples submitted to the stability studies were placed in the temperature control chamber (at 5°C or 25°C). Both chambers are protected from light exposition, being dark places. Therefore, the samples were exposed to the light just for a short time when they were withdrawn for the physical stability analysis.
Line 310 The oiling-off phenomenon is not strictly related to the size of the emulsion in the micrometric range for LAE-99.5.
Response: The authors acknowledge the reviewer’s concern in raising this point. We agree that this statement was improper and removed it from the revised manuscript with other clarification as can be found on Page 8, Line 324-327.
It would be also relevant to discuss the effect of protein and phenolic compounds in relation to saponin content in determining the decrease in interfacial tension and emulsion formation and stability. This is also important since in the conclusion it is reported that: " the emulsifying properties of LAE did not depend only on surface-active compounds and interfacial activities".
Response: Thank you very much for raising this advantage suggestion. In our revised manuscript, we have already included more discussion as can be found on Page 6, Line 273-284.
